# Effect of a Nutritional Intervention on the Intestinal Microbiota of Vertically HIV-Infected Children: The Pediabiota Study

**DOI:** 10.3390/nu12072112

**Published:** 2020-07-16

**Authors:** Talía Sainz, María José Gosalbes, Alba Talavera, Nuria Jimenez-Hernandez, Luis Prieto, Luis Escosa, Sara Guillén, José Tomás Ramos, María Ángeles Muñoz-Fernández, Andrés Moya, Maria Luisa Navarro, María José Mellado, Sergio Serrano-Villar

**Affiliations:** 1Servicio de Pediatría, Hospital Universitario La Paz and IdiPAZ, 28046 Madrid, Spain; luisescosa1983@gmail.com (L.E.); mariajose.mellado@salud.madrid.org (M.J.M.); 2Red de Investigación CoRISpe integrada en la Red en Infectología Pediátrica (RITIP), 28046 Madrid, Spain; prieto_tatoluis_manuel@hotmail.com (L.P.); sguillen@salud.madrid.org (S.G.); josetora@ucm.es (J.T.R.); marisa.navarro.gomez@gmail.com (M.L.N.);; 3Área Genómica y Salud, Fundación para el Fomento de la Investigación Sanitaria y Biomédica (FISABIO), 46010 Valencia, Spain; Maria.Jose.Gosalbes@uv.es (M.J.G.); jimenez-hernandez.nuria@gmail.com (N.J.-H.); andres.moya@uv.es (A.M.); 4CIBER en Epidemiología y Salud Pública, 28029 Madrid, Spain; 5Bioinformatics Unit, Hospital Universitario Ramón y Cajal and IRYCIS, 28034 Madrid, Spain; albatalavera92@gmail.com; 6Servicio de Pediatría, Hospital 12 de Octubre and I+12, 28041 Madrid, Spain; 7Servicio de Pediatría, Hospital de Getafe, 28901 Madrid, Spain; 8Spain Servicio de Pediatría, Hospital Clinico San Carlos and UCM, 28040 Madrid, Spain; 9Laboratorio InmunoBiología Molecular, Sección Inmunología, Hospital General Universitario Gregorio Marañón and Spanish HIV HGM BioBank, Madrid Spain, Networking Research Center on Bioengineering, Biomaterials and Nanomedicine (CIBER-BBN), 28007 Madrid, Spain; mmunoz.hgugm@salud.madrid.org; 10Instituto de Biología Integrativa de Sistemas, Universidad de Valencia, 46003 Valencia, Spain; 11Servicio de Pediatría, Hospital General Universitario Gregorio Marañón, 28007 Madrid, Spain; 12Servicio de Enfermedades Infecciosas, Hospital Universitario Ramón y Cajal and IRYCIS, 28034 Madrid, Spain; serranovillar@gmail.com

**Keywords:** HIV, microbiota, children and adolescents, vertical transmission

## Abstract

Aims: The gut microbiota exerts a critical influence in the immune system. The gut microbiota of human virus immunodeficiency (HIV)-infected children remains barely explored. We aimed to characterize the fecal microbiota in vertically HIV-infected children and to explore the effects of its modulation with a symbiotic nutritional intervention. Methods: a pilot, double blind, randomized placebo-controlled study including HIV-infected children who were randomized to receive a nutritional supplementation including prebiotics and probiotics or placebo for four weeks. HIV-uninfected siblings were recruited as controls. The V3–V4 region of the 16S rRNA gene was sequenced in fecal samples. Results: 22 HIV-infected children on antiretroviral therapy (ART) and with viral load (VL) <50/mL completed the follow-up period. Mean age was 11.4 ± 3.4 years, eight (32%) were male. Their microbiota showed reduced alpha diversity compared to controls and distinct beta diversity at the genus level (Adonis *p* = 0.042). Patients showed decreased abundance of commensals *Faecalibacterium* and an increase in *Prevotella, Akkermansia* and *Escherichia.* The nutritional intervention shaped the microbiota towards the control group, without a clear directionality. Conclusions: Vertical HIV infection is characterized by changes in gut microbiota structure, distinct at the compositional level from the findings reported in adults. A short nutritional intervention attenuated bacterial dysbiosis, without clear changes at the community level. Summary: In a group of 24 vertically HIV-infected children, in comparison to 11 uninfected controls, intestinal dysbiosis was observed despite effective ART. Although not fully effective to restore the microbiota, a short intervention with pre/probiotics attenuated bacterial dysbiosis.

## 1. Introduction

The strong interplay between bacteria and the immune system has been suggested to influence different metabolic and inflammatory conditions, raising the question of whether the microbial communities are new targets for interventions [1]. For reasons that still remain unclear, HIV-infected adults exhibit compositional [2,3] and functional [4,5] changes in the microbiota. The massive depletion of gut-associated lymphoid tissue (GALT) cell populations during acute infection [6], only partially restored by antiretroviral therapy (ART), might well be the main determinant of HIV-related dysbiosis [7]. The extent of GALT injury in the context of vertical Human Immunodeficiency Virus (HIV)-infection has not been studied; the first studies describing the impact, in terms of the composition of a bacterial ecosystem that develops in the presence of an altered gut, have been published recently with controversial results [8,9,10]. There is rationale to support the hypothesis that persistence of thymic function in children may as well have an impact on GALT immune reconstitution [11], and consequently, a distinct effect of vertical HIV infection relative to horizontal HIV infection is plausible

Because it has been described that HIV infection is associated with changes in the microbiota that could play an active role in disease progression, including chronic inflammation [4,12,13] and HIV-vaccine efficacy, [14] some studies have evaluated different dietary interventions in HIV-infected adults with several nutritional products, including prebiotics, probiotics, and synbiotics, among others [15,16,17,18,19]. Collectively, these studies suggest that such strategies could induce some positive immunological effects. Hence, we developed PMT25341, a nutritional supplement designed to improve distinct gastrointestinal defects associated with HIV immunopathogenesis. The aim of this study was to describe the gut microbiota of vertically HIV-infected children, and to address our ability to modify it by means of a nutritional intervention using pre- and probiotics.

## 2. Methods

### 2.1. Study Design

This was a pilot, double blind, randomized, placebo-controlled study. HIV-infected children and adolescents on stable ART were enrolled between October 2013 and November 2014. Cases were HIV-infected patients attending the HIV clinics of four University Hospitals in Madrid, Spain (Hospital La Paz, Clínico San Carlos, Gregorio Marañón y Hospital de Getafe). Inclusion criteria were vertically documented HIV infection, age 6 to 18 years, at least 6 months on stable ART with a regimen containing at least three antiretroviral drugs, HIV RNA suppression and CD4 + T-cell counts ≥350 cells/µL, i.e., with a good immunovirological control. Controls were healthy HIV-uninfected volunteers (siblings of participating children or uninfected children born to HIV-infected mothers from the same clinics, aiming to reach a group of similar age and socioeconomical background) who were recruited as controls for comparison of characterization of the HIV-associated fecal microbiota and did not receive the nutritional product. Exclusion criteria for both groups were the use of concomitant medications; use of systemic antibiotics during the previous three months; and any acute or chronic condition other than chronic HIV infection, including co-infections by hepatitis B or C viruses. Fecal samples from HIV-infected patients were collected at baseline and in the first seven days after a four-weeks course of daily nutritional supplementation. Fecal samples from controls were collected at baseline. Cases were randomized to receive either PMT25341 or placebo.

The study protocol was conformed to the principles or the Declaration of Helsinki and the Good Clinical Practice Guidelines and was approved by the Independent Ethics Committees at all participating Institutions (approval number 173/13). All parents and participants above 12 years of age provided written informed consent/assent.

### 2.2. Nutritional Intervention

Participants were randomized to receive either 20 mg of a specifically designed nutritional supplementation (PMT25341) containing pre/probiotics, oligosaccharides, glutamine, AM3 and vitamin D versus placebo. The composition or PMT25341 is summarized in Table 1 The components included prebiotics (long chain fructo-oligosaccharides, galacto-oligosaccharides) [15]; the probiotic *Saccharomyces boulardii* [17,20]; the essential aminoacids arginine and glutamine [21,22]; a mixture of long-chain fatty acids (eicopentaenoic and docosahexanoic acids, with known anti-inflammatory properties [23]); vitamin D, as a deficiency predicts impaired CD4 recovery; [24] and AM3, a glycopeptide produced by *Ricinus communis,* which promotes the antiviral response of mononuclear cells and mititages LPS-induced inflammation [25]. We used skimmed milk powder as a placebo. The nutritional supplement and the placebo were prepared by Nutricion Médica, S.L. and distributed to the centers in sealed envelopes. Each HIV-infected patient received a box containing sachets with a powder formulation that could be mixed with fruit or beverages, to be administered daily for 4 weeks. Remaining sachets were collected at the follow-up visit to address adherence.

### 2.3. Randomization

The study participants were randomly assigned to PMT25341 or placebo by a computer-generated randomized number system in blocks and were packaged in identically appearing sachets. The study participants and all the researchers involved in the study, including those who handled patient specimens, were blind to the assigned patient group.

### 2.4. Nucleic Acid Purification, Amplification of the 16S rRNA Gene, Sequencing and Bioinformatics Analysis

#### 2.4.1. Nucleic Acid Purification

Fecal samples were collected in sterile tubes with RNA later (Life Technologies) and stored at −80 °C until use. The fecal samples were homogenized, and 5 mL of each sample was diluted with 5 mL of phosphate-buffered saline (PBS). Then, they were centrifuged at 2000 rpm at 4 °C for 2 min to remove fecal debris. The supernatant was centrifuged at 13,000 rpm for 5 min to obtain bacterial pellet. Total DNA was extracted from bacterial pellet with QIAamp DN. Total DNA was quantified with Qubit Fluorometer (ThermoFisher, Waltham, USA) and its integrity verified by standard agarose gel electrophoresis.

#### 2.4.2. Amplification of the 16S rRNA Gene

For each sample, the V3–V4 region of the 16S rRNA gene were amplified from total DNA. The amplicon libraries were constructed following Illumina instructions. The libraries were quantified with Qubit Fluorometer (ThermoFisher, Waltham, USA) and sequenced using the Kit v3 (2 × 230 cycles) in a MiSeq platform (Illumina) at FISABIO Sequencing and Bioinformatics Service, Valencia, Spain. We obtained an average of 79,603 16S rRNA joined sequences per sample. All sequences were deposited in the European Bioinformatics Institute (http://www.ebi.ac.uk/) database (PRJEB35283).

#### 2.4.3. 16S RNA Gene Analysis. Biodiversity and Clustering

The raw reads were processed using QIIME pipeline v1.9 [26]. Taxonomic information on the 16S rDNA sequences were obtained using the Ribosomal Database Project-II (RDP) [27] and the Greengenes database available in the QIIME v1.9 software. We considered only annotations that were obtained with a bootstrap value greater than 0.8, leaving the assignation at the last well-identified level. The Operational Taxonomic Units (OTUs) were generated using UCLUST [28] applying the 97% similarity criterion. To obtain weighted and unweighted UniFrac distance matrices, we first generated a phylogenetic tree and then, applied the core-metrics-phylogenetic method using the QIIME pipeline (https://docs.qiime2.org/2017.12/tutorials/moving-pictures/).

The alpha diversity was determined at OTU level using vegan library from the R package version 3.2.0 (R Development Core Team, 2011). To analyse beta diversity, we applied Principal Coordinates analysis (PCoA) based on weighted and unweighted UniFrac distances. Box plots were generated with in-house R scripts. Nonmetric Multidimensional Scaling (NMDS) was performed in weighted UniFrac matrices using vegan library (function metaMDS) in R package.

We used the linear discriminant analysis (LDA) effect size (LEfSe) algorithm [29] to identify specific taxa biomarkers and reveal significant differences in bacterial abundances between the healthy controls with VIH patients and between the placebo arm and the interventional group.

### 2.5. Statistical Analysis

Mann-Whitney U test was used for the between-group comparisons of continuous variables and Wilcoxon signed-rank matched-pairs test to evaluate between time-points differences in numerical outcomes. To statistically assess the effect of the environmental factors on the bacterial composition, a multivariate analysis of variance based on dissimilarity test (ADONIS) was applied using vegan library from the R package. Statistical analysis was run in Stata v16.0 (StataCorp LP, College Station, TX, USA) and Prism v.7.0, GraphPad, Inc., La Jolla, CA, USA).

## 3. Results

### 3.1. Characteristics of the Study Population

Twenty-four vertically HIV-infected children and adolescents were recruited and compared to 11 control children. Two patients, one in the intervention and one in the placebo arm did not complete the follow-up period and were not included in the analysis of fecal microbiota. Fifteen (62.5%) were female, with a mean age of 12 ± 3.9 years. All were on ART and virologically suppressed. The main characteristics of controls, and placebo vs. interventional group are shown in Table 2.

### 3.2. Analysis of Fecal Microbiota Structure

Alpha diversity is used to measure the richness and evenness of bacterial taxa within a community. Contrary to what is usually reported in many disease states, HIV-infected subjects displayed a similar richness of species (Figure 1A). At the genus level, we did not observe differences on richness indicators Chao1 and ACE (*P* = NS. for all comparisons, Figure 1B,C) between HIV-infected children and controls at baseline.

Beta diversity focuses on the ecological distances between samples. Analysis of weighted Unifrac distances, which quantifies the compositional dissimilarity between samples based on the relative abundance of taxa and their phylogenetic relatedness revealed significant differences between the microbiota structure of controls compared to the HIV-infected children (Adonis *p* = 0.042), which lost the statistical significance after the nutritional intervention (Adonis *p* = 0.272), suggesting that PMT25341 attenuated the imbalance on the microbial composition associated with the HIV infection (Figure 2A). In addition, weighted Unifrac distances were compared using Nonmetric Multidimensional Scaling (NMDS) analysis, a method to visualize similarities or dissimilarities in high-dimensional data. This analysis revealed two different clusters based on the HIV serostatus (Figure 2B, *p* = 0.059), but when baseline vs. after treatment samples were compared, no clear clusters were identified (Figure 2C, *p* = 0.797)**.**

Among the most predominant bacteria at the genus level, differences between the HIV-infected vs. the control groups could be appreciated in the visual inspection of the compositional bar plots (Figure 3). HIV-infected children showed enrichment for *Prevotella*, *Akkermansia* and *Escherichia*, and depletion of *Faecalibacterium and Bifidobacterium.* Other predominant genera included *Roseburia, Parabacteroides and Sutterella.* No clear effects attributable to the intervention were identified.

### 3.3. Bacterial Biomarkers of HIV Infection and Taxa Significantly Affected by the Intervention

We used the linear discriminative analysis (LDA) effect size (LEfSe) biomarker discovery tool, a method that addresses the challenge of finding microorganisms that consistently explain the differences between two or more microbial communities, to elucidate which genera were driving divergence between both groups (Figure 4). In the comparison of basal samples from HIV-infected children vs. healthy controls, we found that HIV infection drives enrichment of the pathobionts *Fusobacterium* and *Coprobacillus*, while the symbionts *Faecalibacterium, Lachnospira, Dorea, Clostridium* and *Lactococcus* were significantly depleted. After the intervention, LEfSe analysis did not detect significant taxa associated with PMT25341.

## 4. Discussion

In this study, we found that the gut microbiota of vertically HIV-infected children is distinct from that of healthy controls. Alpha diversity, a measure of the bacterial richness and evenness within a community, was significantly lower in the HIV-infected children, although the characteristics at a compositional level differ from the findings described in HIV-infected adults.

Although there is controversy regarding the effects of HIV, both in children and adults, it is generally assumed that a lower alpha diversity is reflective of disease states. Our findings at the compositional level agree with the widely accepted assumption that changes in the microbiota associated with disease states might fuel pathogenic circles. The most clearly depleted genus in the HIV-infected populations was *Faecalibacterium.* This genus is a predominant commensal in the healthy gut and exerts an important role in inducing regulatory T cells [30] and decreasing intestinal permeability [31]. Of note, this genus is also depleted in other inflammatory conditions, such as Crohn’s disease and ulcerative colitis [32,33]. In keep with studies in HIV-infected adults, we also found a significant depletion of the major butyrate-producers *Faecalibacterium* and *Lachnospira.* As described in a recent study in perinatally HIV-infected children [9], we also observed enrichment for *Prevotella* in HIV-infected children, a microbiota trait that is being shown to exert systemic pro-inflammatory effects [34]. This is of interest as the shift from *Bacteroides* to *Prevotella* predominance classically associated to HIV infection in adults is now interpreted as an enterotype characteristic of men who have sex with men rather than determined by HIV infection [35].

However, some findings in our study contrast with previous data in children and adults. *Dorea,* a genus that we found positively correlated with immunosenescence in the fecal microbiota of HIV-infected adults [16] was decreased in the HIV-infected children. *Akkermansia* is a key taxa able to stimulate the production of human cytokines, which exhibit functional redundancy with other genus including *Sutterella, Akkermansia, Bifidobacterium, Roseburia* and *Faecalibacterium prausnitzii*, typically associated with beneficial functions for the host [36,37]. We think that the observed increase of *Akkermansia* with respect to controls (Figure 3) might represent a compensatory adaptation to a chronic perturbation (i.e., HIV infection) to mitigate the harm in its habitat due the overlapping functions with *Faecalibacterium* [36,37]. LefSe analysis also indicated that HIV-infected children were enriched with *Fusobacterium*, a well-known pathobiont associated with diseases states. However, we also found enrichment in *Coprobacillus*, a gram positive anaerobe in the Eysipelotrichaceae family, a taxa consistently typically enriched in HIV-infected adults, which we found specifically represented in the fraction of active bacteria among HIV-infected subjects with greater ART-mediated immune recovery in a study using meta-proteomics [4].

In addition, this is the first study exploring the impact of a nutritional intervention on the microbiota in vertically HIV-infected patients. While the intervention attenuated HIV-associated dysbiosis at the beta diversity level, we did not observe significant changes at the alpha diversity level. Prebiotics have been previously evaluated in HIV-infected adults, showing a decrease of the immune activation markers sCD14 and in CD4^+^CD25^+^T-cells in ART-naïve patients [15]. Other studies have combined probiotics, prebiotics and other compounds to target the diverse immunological deficits in the gut [19]. We previously used a combination of prebiotics and glutamine in a pilot-controlled study, showing modest effects on T-cell activation and fecal microbiota structure, especially in ART-naïve individuals [38]. PMT25341 included components previously associated with an amelioration of gut epithelial barrier integrity, decline of bacterial translocation and associated with immune recovery [15,17,20,21,22,23,24,25]. We have previously reported the effects of PMT25341 in a placebo-controlled trial in HIV-infected adults diagnosed with less than 350 CD4^+^ T-cell counts/uL [39]. In contrast with some exploratory studies with nutritional interventions in ART-naïve or adults on stable triple ART where nutritional interventions have shown a modest impact on some immunological outcomes [15,17,19,38,40], we did not detect evidence of clear effects on immune recovery or gut microbiota structure in late presenters. We think that the fact that the same product in children was able to attenuate the HIV-associated changes on fecal microbiota supports the notion that the microbiota is less resilient in children than in adults and a better scenario to assess interventions aimed at shaping the microbiota.

It is recognized now that the assemblage of bacterial species before achieving a stable composition occurs progressively during the first years of life. The microbiome is likely to achieve a stable state during adulthood, yet the landscapes of stable states of the human microbiome is still unknown [41,42]. From an ecological perspective, microbiota-targeted interventions can be expected to lead to stronger effects when applied in microbial communities who have not achieved yet a stable configuration [41]. On top of that, our study allows us to interrogate the impact of HIV on the gut microbiota in a population in which major confounders of the microbiota, such us smoking [43], sexually transmitted diseases [44] or sexual orientation [35], a major methodological hurdle in studies in HIV-infected adults, are absent, and in the presence of a carefully selected control group. The main limitation of our study is the limited sample size, which warrants caution in the interpretation of the results, to be confirmed in larger studies. The duration of the intervention might have limited our ability as well to find any effect, as interventions are usually longer in most studies published addressing the impact of nutritional supplementation. Finally, although we did not evaluate the dietary habits, which could have influenced the results, although we preferentially selected as controls HIV-uninfected born from the same mother with HIV.

## 5. Conclusions

This is the first study characterizing the effects of a short dietary intervention on the microbiota in vertically HIV-infected children. We found that vertical HIV infection is associated with changes in gut microbial communities, which were attenuated after a 4-week nutritional intervention, without a clear impact at the community level. The analysis of clinical endpoints, including effects on immune recovery, vaccine response or immunoactivation are ongoing, to address whether this is a useful approach to improve the health of children living with HIV.

## Figures and Tables

**Figure 1 nutrients-12-02112-f001:**
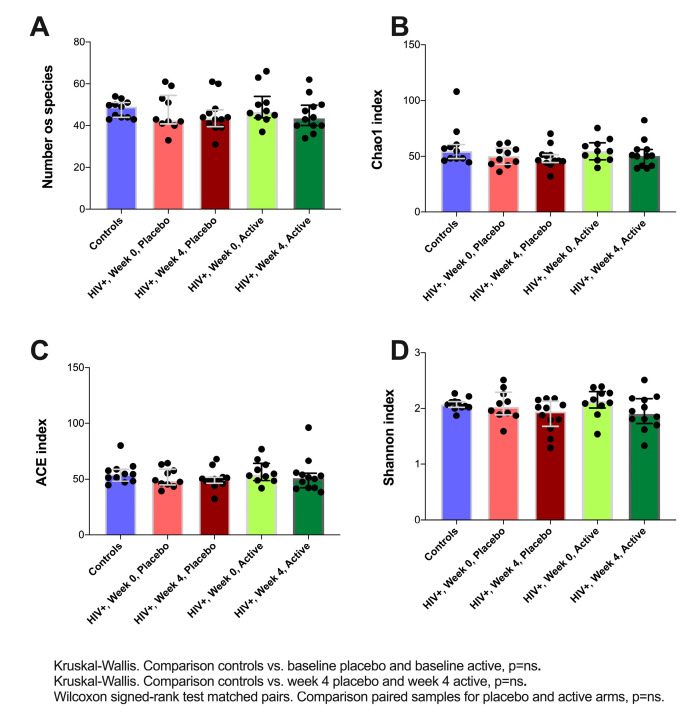
Alpha diversity metrics comparison across study groups. (**A**), Number of species, illustrates de number of unique taxa identified. (**B**), Chao1, an estimation of real richness based on the number of taxa supported by either 1 or 2 sequencing reads. (**C**), ACE, an estimation of real richness based on the number of taxa supported by 10 reads. (**D**), Shannon, a metric to quantify the uncertainty of the taxonomic identity of an unknown entity in a community based on the number of different entities and their abundance.

**Figure 2 nutrients-12-02112-f002:**
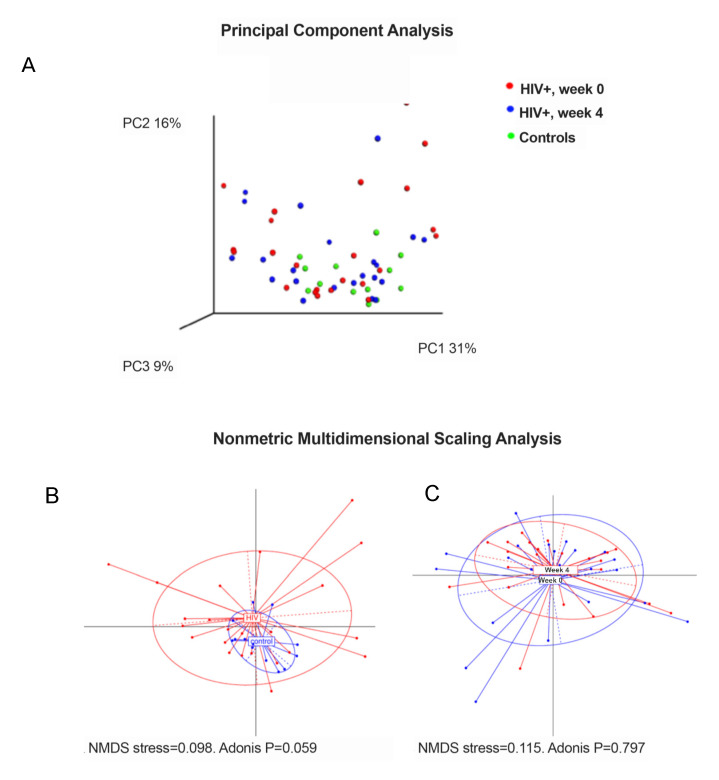
Beta diversity analysis based on weighted Unifrac distances. (**A**), Principal component analysis based on weighted Unifrac distances. HIV-infected children at baseline vs. controls, Adonis *p* = 0.042; HIV-infected children at week 4 vs. controls, *p* = 0.272; HIV-infected children at week 0 vs. week 4, *p* = 0.729. (**B**) Nonmetric multidimensional scaling analysis, HIV-infected children at baseline vs. controls and (**C**) HIV-infected children at baseline vs. week 4. NMDS, Nonmetric Multidimensional Scaling.

**Figure 3 nutrients-12-02112-f003:**
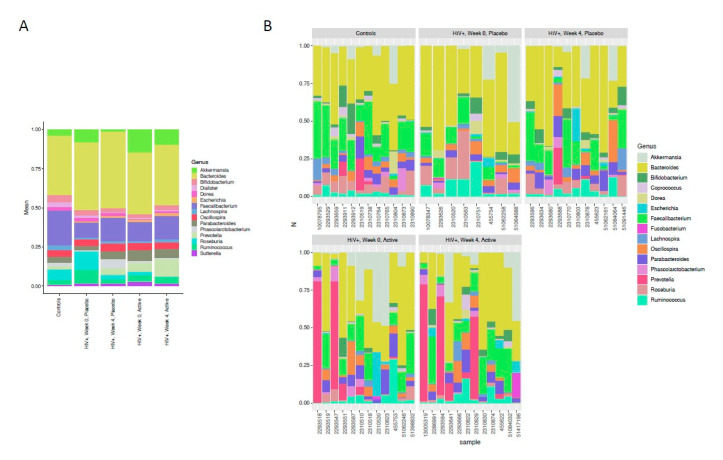
Relative abundance of top 15 more prevalent genus. Differences between the HIV-infected vs. the control groups could be appreciated, without clear differences related to the intervention. Figure 3 (**A**) shows the mean relative abundances of the top 15 most abundance genus in each group. Figure 3 (**B**) shows the individual relative abundances of each study subject at each timepoint.

**Figure 4 nutrients-12-02112-f004:**
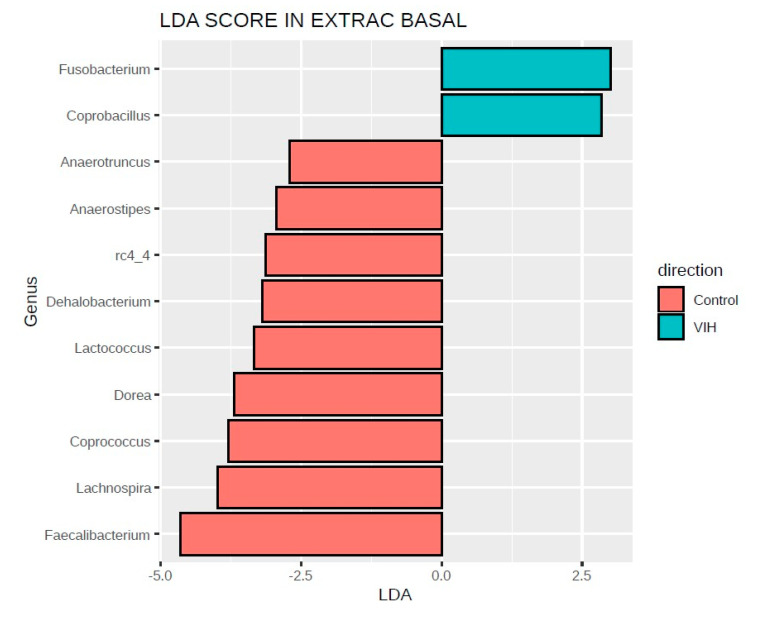
Bacterial taxa driving divergences between groups using LEfSe analysis, HIV-infected children vs. controls at baseline. The analysis week 0 vs. week 4 samples from HIV-infected subjects in the PMT25341 arm did not identify any significant change in the abundance of bacterial genus. The linear discriminative analysis (LDA) scores (log 10) for the most prevalent taxa among controls, represented in red, are plotted in the positive scale, whereas LDA-negative scores indicate those taxa enriched in the HIV-infected group at baseline or after the intervention are represented in green.

**Table 1 nutrients-12-02112-t001:** Product composition: Intake per day.

	Active	Placebo
Energy	65.3 kcal	74.9 kcal
Total lipids	5.1 g	0
EPA	3.2 g	0
DHA	0.3 g	0
GLA	1.7 g	0
Total carbohydrates	10.3 g	10.9 g
Long-chain FOS	4 g	0
Short-chain GOS	3.3 g	0
Maltodextrin (excipient)	3	0
Total proteins	3.3 g	6.9 g
L-Glutamine	2 g	0
L-Arginine	1.2 g	0
Others	1.2 g	0
*Saccharomyces boulardii*	0.17 g	0
AM3	1 g	0
Vitamin D	5.3 × 10^3^ MU	0
TOTAL	21.1	21.1

Abbreviations: EPA, eicosapentaenoic acid; DHA, docosahexaenoic acid; GLA, γ-linolenic acid; FOS, fructo oligosaccharides; GOS, galacto oligosaccharides.

**Table 2 nutrients-12-02112-t002:** Main characteristics of the study cohorts at baseline.

	Controls *N* = 11	Placebo *N* = 12	Nutritional Intervention *N* = 12	*p*
Female (*n*,%)	4 (36.4)	7 (58.3)	8 (66.7)	1.000
Age (years), mean (SD)	10 (4.4)	13.8 (3.6)	10 (3.4)	0.064
Caucasian (*n*,%)	7 (64)	6 (50)	7 (58.3)	1.000
CD4 count (cells/mm^3^)	-	556 (453–754)	852 (617–1182)	0.0496
CD4/CD8 ratio	-	1.1 (0.56–1.67)	1.4 (1.09–1.94)	0.106
CD4 Nadir (cells/mm^3^)	-	333 (169–382)	519 (384–979)	0.006
PI based ART (*n*,%)	-	8 (66.7)	9 (75)	0.136

All values are expressed in median (IQR) except otherwise specified. ART: antiretroviral treatment. PI: protease inhibitor.

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
