# Peer review of "Effect of a Nutritional Intervention on the Intestinal Microbiota of Vertically HIV-Infected Children: The Pediabiota Study"

_nutrients, 2020, doi:10.3390/nu12072112_

Round 1

Reviewer 1 Report

An interesting well-designed clinical study assessing a nutritional intervention with a predetermined supplement in children with vertically transmitted HIV. Since this is a key point of the study design it is suggested that supplementary table depicting the details of the content of the diet supplement be part of the publication rather than a suppletory data. Add a short explanation of why each of the components.

Please explain if the general diet of the children in the study was assessed and compared?

Add a reference in the introduction line 50 after the intervention

Add reference for vertical infection line 54.

These publications aim to reach a wide range of readers who may not be aware of the clinical definitions.

Abbreviations should be explained when first used. Please revise.

At the end of the introduction please explain which were the expected benefits of the intervention or why is important to assess the difference in microbiota composition within the groups tested and why to modify it?

Did the control children receive the supplementation?

Review spacing between headings and subheadings to avoid confusion and facilitate reading.

Revise the section of funding it appears there is a repetition of paragraphs.

The reference section needs spacing.

Since there are so many co-authors could mark with an asterisk relevant publication from the group to substantiate and support the previous experiences with the methodology?

Author Response

We thank the editorial team and the reviewers for the valuable input that has contributed to significantly improve the manuscript.

RESPONSES TO REVIEWER 1

An interesting well-designed clinical study assessing a nutritional intervention with a predetermined supplement in children with vertically transmitted HIV. Since this is a key point of the study design it is suggested that supplementary table depicting the details of the content of the diet supplement be part of the publication rather than a suppletory data. Add a short explanation of why each of the components.

Author: following this Reviewer`s suggestion, a reference has been described each component in the method with a brief rationale to justify the inclusion of each component. In addition, we have moved the table detailing the composition from the supplemental materials to the main text.

Please explain if the general diet of the children in the study was assessed and compared?

Author: Unfortunately, we did not assess the dietary level. We addressed this potential bias in the study by recruiting preferentially cohabiting HIV-uninfected siblings born from the same HIV-infected mother, and hence, under a similar dietary pattern, or, when this was not possible, from children born from other mothers with HIV from the same clinic.

To give consideration to this Reviewer’s comment, we have added the following text in the Discussion, section of Limitations:

“Finally, although we did not evaluate the dietary habits, which could have influenced the results, although we preferentially selected as controls HIV-uninfected born from the same mother with HIV.”

Add a reference in the introduction line 50 after the intervention

Author: A reference has been added.

Add reference for vertical infection line 54.

This line states “the extent of GALT injury in the context of vertical HIV-infection has not been studied”. We have performed a literature review and found no original articles to cite. This can be explained by the ethical issues inherent to performing endoscopic studies with tissue sampling in children for research. Hence, because there are no studies to reference of the histological effect of HIV on GALT in pediatric populations, no references can be added here.

The following sentence stating “…first studies describing the impact in terms of composition of a bacterial ecosystem that develops already in the presence of an altered gut have been published recently with controversial results” is referenced with cites #8-10.

These publications aim to reach a wide range of readers who may not be aware of the clinical definitions.

The acronyms provided in the introduction have been spelt-out and we have reviewed the manuscript bearing in mind that the audience will essentially be from outside the HIV field.

Abbreviations should be explained when first used. Please revise.

We have spelt out now the acronyms in the introduction: GALT; HIV; ART.

At the end of the introduction please explain which were the expected benefits of the intervention or why is important to assess the difference in microbiota composition within the groups tested and why to modify it?

RESPONSE: to account for this Reviewer’s comment, we have modified the second paragraph in the discussion and explained the rational to attempt to modify the microbiota of vertically HIV-infected children.

Did the control children receive the supplementation?

RESPONSE: controls were used as the reference to characterize the microbial signature associated with HIV. Only HIV-infected children received the supplementation.

We have clarified this point in the methods.

Review spacing between headings and subheadings to avoid confusion and facilitate reading.

We have followed this instruction.

Revise the section of funding it appears there is a repetition of paragraphs.

We have reviewed the funding section (lines 402-411)

The reference section needs spacing.

Spacing has been changed by the editorial team.

Since there are so many co-authors could mark with an asterisk relevant publication from the group to substantiate and support the previous experiences with the methodology?

We have highlighted with an asterisk such references in the bibliography section.

Reviewer 2 Report

The authors describe in the manuscript the influence of prebiotics and probiotics on intestinal colonization. Prebiotics are given. Probiotics are neither defined nor defined in their composition.

In the study design, the reference to the registration number of the ethics opinion is missing.

The microbial analysis was carried out using 16s RNA analysis. This is well described. The statistical analysis does not show any differences between the groups. Do the microorganisms found in the intestinal flora belong to the applied probiotics?

Unfortunately, the effect of the intervention are minor. 

Author Response

We thank the editorial team and the reviewers for the valuable input that has contributed to significantly improve the manuscript.

---------------------

RESPONSES TO REVIEWER 2

The authors describe in the manuscript the influence of prebiotics and probiotics on intestinal colonization. Prebiotics are given. Probiotics are neither defined nor defined in their composition.

Author: following this Reviewer`s suggestion, a reference has been described each component in the method with a brief rationale to justify the inclusion of each component (lines 130-137). In addition, we have moved the table detailing the composition from the supplemental materials to the main text (line 141).

In the study design, the reference to the registration number of the ethics opinion is missing.

Author: we have included the approval number in the Ethics Committee statement in the methods (line 91).

The microbial analysis was carried out using 16s RNA analysis. This is well described. The statistical analysis does not show any differences between the groups. Do the microorganisms found in the intestinal flora belong to the applied probiotics?

Author: the probiotic used is a yeast (Saccharomyces boulardii). Because we the amplicons sequenced the were amplified from the bacterial biomarker, the 16s RNA gene, we could not perform such analysis.

Unfortunately, the effect of the intervention are minor.

Author: We agree with the Reviewer that the effects were marginal. This is, however, an important message for the field, and is consistent with the findings obtained already published from our group using the same nutritional supplement in HIV-infected adults who received the supplement for 48 weeks (PMID: 29788075). The emerging body of evidence in this population suggests that the microbiota in HIV-infected individuals is resilient to such nutritional interventions.

Round 2

Reviewer 2 Report

The manuscript has been modified as suggested by the reviewers. It shows well trhe results of this dietary study and is now easily to follow.